# An Assessment of Seasonal Differences in Fish Populations in Laizhou Bay Using Environmental DNA and Conventional Resource Survey Techniques

**Shuqin Dai [1,2]**, **Maojuan Bai [1]**, **Hui Jia [2,3]**, **Weiwei Xian [2]** and **Hui Zhang [2,4,*]**

1   College of Environment and Safety Engineering, Qingdao University of Science and Technology, Qingdao 266000, China
2   CAS Key Laboratory of Marine Ecology and Environmental Sciences, Institute of Oceanology, Chinese Academy of Sciences, Qingdao 266071, China
3   School of Marine Sciences, Ningbo University, Ningbo 315211, China
4   University of Chinese Academy of Sciences, Beijing 100049, China
*   Correspondence: zhanghui@qdio.ac.cn; Tel.: +86-0532-8289-1860

**Abstract:** In recent years, environmental DNA (eDNA) technology has gradually improved, and it has been increasingly used to monitor marine fish. The decline and seasonal fluctuations of fish resources in Laizhou Bay, Bohai were studied using eDNA technology and compared with the results of conventional fish resource survey methods. In November 2020 (autumn), March 2021 (spring), and July 2021 (summer), 12 samples were collected each quarter in Laizhou Bay and adjacent waters for a total of 36 eDNA samples, and 47 fish species were identified. During the same trip, ground cages, gillnets, and trawls were used during two seasons. Fishery resource surveys were conducted at 12 sites from November 2020 (autumn) to March 2021 (spring), and in total 11 fish species were found. Our study found that fishery resources in Laizhou Bay significantly fluctuated with seasonal changes. Additionally, compared with traditional surveys, eDNA information included the same results, but also included fish that could not be collected because of the technical limitations of traditional surveys. Therefore, this study provides more accurate seasonal information for fish in Laizhou Bay, which is of great significance for the long-term management and conservation of coastal biodiversity.

**Keywords:** Laizhou Bay; eDNA; fish diversity; seasonal variation; conventional resource survey





## 1. Introduction

Laizhou Bay, the sea area south of the mouth of the Yellow River to Longkou, is the largest harbor in Shandong Province and one of the three largest harbors in the Bohai Sea. Laizhou Bay provides essential spawning and nursery plant communities for numerous fishery organisms, including commercial species such as Chinese shrimp (*Fenneropenaeus chinensis*) and large yellow croaker (*Larimichthys polyactis*) [1]. Laizhou Bay is under extreme stress because of excessive fishing pressure and changes in the environment [2]. Under the influence of human activities, the number of dominant fish species in Laizhou Bay is declining, shoreline erosion is severe, and the area of seawater intrusion is annually increasing [3]. The use and preservation of Laizhou Bay in a sustainable manner have become subjects of interest for fisheries resource management [1]. Moreover, understanding and monitoring Laizhou Bay fish resources is essential for estuarine conservation. Because of the diversity of marine fish species and alterations in the aquatic environment, their occurrence and distribution are not completely understood. Comprehensive ecological monitoring is technically challenging, and conventional monitoring and survey methods may be harmful to fishes [4].

Environmental DNA (eDNA) is used to identify the mixture of genomes found in environmental samples from various organisms that are highly persistent in the natural

environment. DNA fragments that are shed from organisms, such as from their skin, feces, saliva, gametes, and secretions, are directly extracted from environmental samples (water, soil, air, and soil particles) and then examined by PCR amplification and high-throughput sequencing [5]. The genomic DNA is used to obtain species information. eDNA technology eliminates the need to isolate species and reduces the environmental impact of biomonitoring [6]. In 1987, Ogram et al. [7] first introduced the concept of eDNA by studying the DNA extracted from sediments. In 1990, Lee and Furman [8] first used e DNA technology to study the microbial diversity in seawater. Several studies have laid a theoretical and empirical foundation for the application of eDNA technology in Marine fish monitoring, and it was not applied to aquatic organisms until 2008 [9]. In 2012, Minamoto et al. [10] monitored fish species composition by using eDNA, showing that the information obtained from eDNA was not significantly different from traditional methods; they concluded that eDNA technology was more time-saving and labor-saving. The development of eDNA provides a new means for ecological and biodiversity monitoring. In particular, the technology makes fish surveys more energy-efficient, cost-effective, and non-invasive, all of which have facilitated its widespread adoption. Many studies have shown that eDNA technology has also become a method to evaluate the fragmentation of large biological communities in marine ecosystems [11,12]. In recent years, eDNA technology has been proven to be effective in monitoring rare, endangered, protected and invasive species in other applications, such as assessing biodiversity and determining species composition, population trends, and ecological functions. Especially in the past two years, more and more researchers used eDNA technology to investigate fish resources. For example, Sigsgaard et al. have successfully monitored the seasonal changes of fish community structure along the coast of Denmark by using 12s rRNA genetic markers, and this also provides a research basis for this study [13]. In 2018, Michael et al. [14] used eDNA technology to monitor marine protected areas in Australia and obtained 13 orders and 82 species of fish. eDNA's monitoring of fish communities extends far beyond offshore waters. In 2020, Beverly and his team used eDNA technology to monitor fish communities in the deep sea [15]. They concluded that eDNA technology can detect the largest number of fish communities compared with other technologies. Not only that, our team has successfully used eDNA technology to monitor fish diversity in both the South China Sea and the East China Sea [16,17]. In China's Bohai Sea (Laizhou Bay), the traditional net fishing method is always used to monitor the fish community, and the method is obviously not feasible if you want to monitor the fish community during the fishing ban period [4]. As one of the three major ports in the Bohai Sea and an important aquaculture base in Shandong, Laizhou Bay provides a place for habitat and reproduction for many wild economic fish species [1]. In recent years, Laizhou Bay is facing many ecological tests, such as the destruction of coastal wetlands, marine environmental pollution and overfishing, and these have had a great impact on marine fish [18]. Therefore, it is imperative to monitor marine fish in order to protect the marine ecological environment. Moreover, in this study, it is the first time we used eDNA technology to monitor the fish community in Laizhou Bay.

The purpose of this study was to: (1) use eDNA to examine fish diversity and seasonal fluctuations in Laizhou Bay; (2) compare conventional survey techniques and eDNA for fish resource monitoring in Laizhou Bay; and (3) use statistics of fish composition in the region during 2020–2021 to provide a theoretical foundation for fish resource management in this region. Although eDNA technology has many advantages, it cannot completely replace traditional methods. Therefore, this study combines the eDNA technology with traditional technologies for study. eDNA complements the conventional survey techniques for studying fish resources in Laizhou Bay and provides a more comprehensive understanding of the composition of fish species in Laizhou Bay. This is the first time that eDNA has been used to determine the species composition of fish in Laizhou Bay. These findings provide an effective monitoring system for the management of fishery resources in Laizhou Bay.

## 2. Materials and Methods

### 2.1. Research Sites

In total, 36 samples were collected for eDNA analysis from Laizhou Bay and its waterways in November 2020 (autumn), March 2021 (spring time), and July 2021 (summer time), with 12 samples from each season collected on the same day and immediately preserved in preservation solution. There are 12 stations (Figure 1), of which 1, 2, and 3 belong to the marine ranch area; 4, 5, and 6 to the reef area; 7, 8, and 9 to the surrounding reef area; and 10, 11, and 12 to the control area (Table 1). (Note: The division of sampling stations is divided by shipping companies to facilitate the sampling of ships, so the sub-areas divided are not related to this study.)

**Table 1.** Sampling station coordinates.

| Sub-Area | Station | Longitude | Latitude |
|---|---|---|---|
| Area for the integration of wind power in pastures | 1 | 119°39.850 | 37°14.963 |
| | 2 | 119°41.092 | 37°15.071 |
| | 3 | 119°42.169 | 37°15.031 |
| Reef area | 4 | 119°41.050 | 37°16.623 |
| | 5 | 119°40.853 | 37°18.382 |
| | 6 | 119°40.154 | 37°20.442 |
| Surrounding reef area | 7 | 119°42.300 | 37°18.300 |
| | 8 | 119°37.889 | 37°20.683 |
| | 9 | 119°42.441 | 37°20.974 |
| Control area | 10 | 119°34.897 | 37°16.623 |
| | 11 | 119°34.398 | 37°18.224 |
| | 12 | 119°34.103 | 37°20.153 |

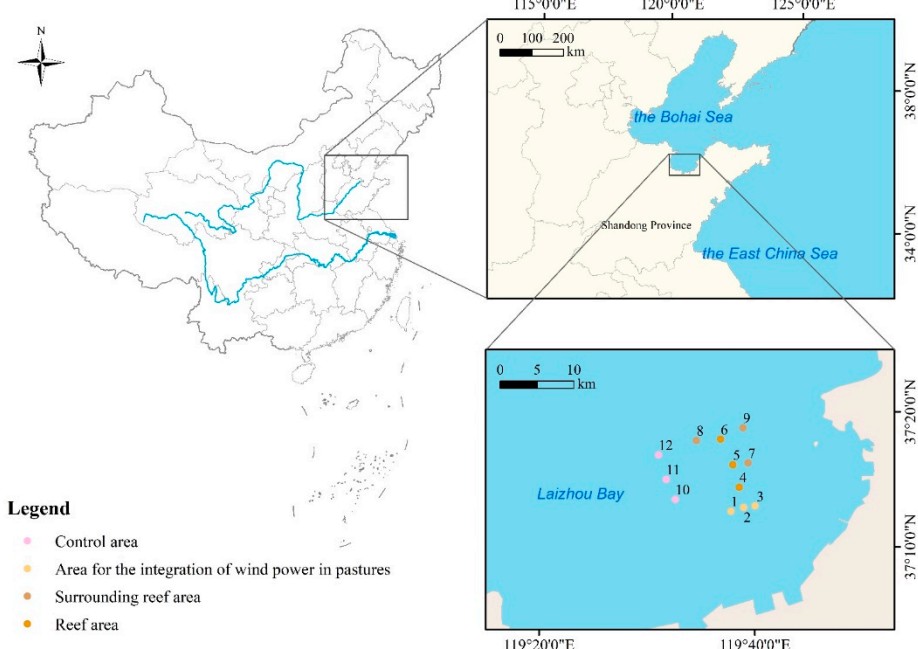

**Figure 1.** Station distribution map (the right side shows the specific distribution of stations in Laizhou Bay).

### 2.2. Sampling

#### 2.2.1. eDNA Sampling

To prevent contamination of the water samples, the sampling equipment was thoroughly cleaned before being re-used after the sampling period. Then, 1–2 L water samples were taken from the surface water layer based on marine measurement specifications, and

Urine DNALOCKER (Tiandz Inc., Beijing, China) was added to the water samples in a 1:10 ratio in brown closed jars to prevent DNA degradation during the transportation of water samples. Before the sample is filtered, we shake the water sample in the sampling bottle and measure 1L water sample with a measuring cylinder for filtration. Through this process control, the water sample will be strictly controlled at 1L. The water samples were immediately filtered when transported to the laboratory using a TM-GP filter unit; (pore size, 0.22 μm; EMD Millipore Corp, Billerica, MA, USA) for greater DNA enrichment, each sample bottle was filtered two times. The water was then emptied from the extraction column using a hypodermic needle and deposited in a DNA maintenance buffer (Tiandz Inc., Beijing, China). As negative controls, equal volumes of filtered deionized water were used to limit experimental error and confirm the absence of contaminants. Original samples were therefore stored in a refrigerator at a temperature of $-20$ °C.

### 2.2.2. Conventional Sampling Techniques

Conventional fishing techniques, including ground cages and gillnets for aquatic organisms such as fish and crustaceans, were used at the same stations where eDNA sampling occurred. We combined fish information from only two seasons for comparison with eDNA data (Laizhou Bay was closed to fishing during summer 2021).

The survey was conducted between November 2020 (autumn) and March 2021 (spring). The 12 stations were surveyed for their fishery resources using ground cages, gillnets, and trawls. The mesh size of the ground cage was 2 cm; one string consisted of 24 sections, and three strings of ground cage were grouped together. The gillnets were 50 m with mesh sizes of 40 mm, 50 mm, and 60 mm, for a total length of 150 m and a net height of 1.2 m. The placement duration for both the ground cage and gillnet was 48 h. The sea area surveyed was 10,795 ha, and the reef area covers an area of about 3400 ha. In addition to identifying the fish species in the catches, the weight and number of tails of each fish were recorded. The catch per unit time (kg/h) and the number of tails per unit time (tails/h) were used to calculate the average catch per unit time.

### 2.3. eDNA Analysis Methods

### 2.3.1. eDNA Extraction, Library Construction, and Sequencing

Raw samples that were frozen with dry ice were sent to Personal Biology (218 Yindu Road, Xuhui District, Shanghai, China) for processing. The Sterivex filtering units were extracted by Miya et al. [19] using the DNeasy Blood and Tissue Kit (Qiagen, Hildesheim, Germany). A fish eDNA universal primer pair (MiFish-U/E) was used for PCR amplification [19]. The acceptable length of the amplified PCR product (180 bp) for high-throughput sequencing systems makes MiFish-U/E an attractive primer pair for ambient DNA metabarcoding monitoring approaches for fish. Using 1.2% agarose gel electrophoresis, the DNA extraction quality was evaluated. The Quant-iT PicoGreen dsDNA Assay Kit was used to quantify PCR products on a microplate reader (BioTek, FL×800), which were then combined based on the amount of data required for each sample. Illumina's TruSeq Nano DNA LT Library Prep Kit was used to construct the libraries. To retrieve more species information, we analyzed each sample three times and mixed all PCR products in equal volume. Meanwhile, a negative control was established to detect microbial contamination from the environment or reagents. Any negative control group with amplified bands was not used for subsequent experiments.

Subsequently, 1 μL of library DNA was extracted and analyzed for quality control on an Agilent Bio-analyzer machine via the Agilent High Sensitivity DNA Pack 2100; an approved library should have a single peak and no intersection. Using the Quant-iT PicoGreen dsDNA Assay Kit on a Promega QuantiFluor, the concentration of competent libraries was calculated to be at least 2 nM. We performed 2250-bp paired-end sequencing on libraries using an Illumina NovaSeq machine and a NovaSeq 6000 SP Reagent Pack (500 cycles). The libraries were dissolved to 2 nM in a gradient before being proportionally blended to the required amount of data. For sequencing, the mixed libraries were denatured

to a single strand with 0.1 N NaOH. The quantity of library to be sequenced varied between 15 and 18 pM.

### 2.3.2. Pre-Processing and Information Quality Control

Illumina technology was employed in this experiment for paired-end sequencing of community DNA fragments. The raw high-throughput sequencing data were evaluated for sequence quality. These sequences were then segmented using indexing and barcode data, and barcode sequences were eliminated [16]. Raw sequencing data were stored in FASTQ format. To determine the location of the eDNA fragment on the genome or gene, the sequencing reads in the FASTQ file were read against a specific reference genome.

Before proceeding with sequence alignment, reads were evaluated to ensure they were of sufficiently high quality to assure the accuracy of subsequent analyses. Sequencing quality control was carried out as follows: sequencing junctions and primer sequences were removed, and poor-quality information was filtered to ensure information quality using DADA2 [20], which is a software program that primarily handles primer removal, quality filtering, denoise, splicing, and chimera removal. Using the analysis software QIIME2, the following steps were followed: first, the QIIME cutadapt trim-paired tool was used to excise the primer fragment of the sequence and discard the sequence of unmatched primers; DADA2 was then called using DADA2 via the QIIME DADA2 denoise-paired command for quality control, denoising, stitching, and dechimerization. The above steps were separately conducted for each library. After denoising all libraries, the amplicon sequence variation (ASV) feature sequences and ASV table were merged, and the singleton ASVs were removed. Sequence Length Distribution Statistics, which uses an R language script, was used to count the length distributions of high-quality sequences contained in all samples. It no longer clusters by similarity, but merely by dereplication, which is the same as clustering by 100% similarity. Each dereplicated sequence produced following DADA2 quality control was referred to as an ASV. Clean information refers to the high-quality reads or bases received following the preceding sequence of quality controls, and clean data are also available in FASTQ format. The total numbers of high-quality reads in the three quarters after data processing were 1,479,033 in autumn 2020, 941,425 in spring 2021, and 1,006,748 in summer 2021. It is found in the study of Damien et al. [21] that there is a positive correlation between the sequence reading of eDNA and species abundance, but the two are not equal signs. On this basis, the sequence readings were used as the criteria for identifying dominant species. In terms of diversity analysis, the fish species identification and species composition heat map analysis, this paper has selected those species with the top 22 sequence readings for further analysis to exclude the false positive of eDNA. At the same time, the analysis of these aspects is also based on the positive correlation between sequence reading and species abundance. Although there are many factors that affect the detection rate of eDNA in the marine environment, such factors that cause the error of the results are still within the controllable range after the team's repeated experiments and the control of the amount of sequence reading [16].

### 2.3.3. Identification of Fish Species

Following the acquisition of high-quality sequence distributions, we used the NCBI Basic Local Alignment Search tool to check all portions that matched fish and conducted a blast search for comparison. For annotation screening, QIIME2 was used, and the final results were compared with MitoFish and NCBI's Genome database [16]. The comparison criteria were as follows: identity > 95% and E-value = $10^{-5}$ [22]. For the final results, the best comparison was selected for each cluster. QIIME2's classify-sklearn plug-in was employed [23]: species annotation was conducted in the QIIME2 program using default settings and a pre-trained Naive Bayes classifier was used for each ASV feature sequence.

After the above process, the species were roughly annotated, and the basic species annotation results were obtained. Then, the likely deviation of results caused by experimental pollution was removed from the annotated outcomes based on the real state of fish resources

in Laizhou Bay. We used Fishbase (https://www.fishbase.se/search.php; accessed on 26 March 2022) and consulted books about fishery resources in the Laizhou Bay to confirm if they were local. Finally, the results of the precise species annotation were obtained [24]. We chose the 22 most common species for further seasonal biomarker research based on the information base building of fish survey information and the taxonomically plentiful supply in Laizhou Bay because the majority of species were only detected in summer. QIIME2 software was used for Random Forest algorithm analysis. Based on the absolute abundance table of a taxon at the species level, the function "classify_samples_ncv" in the Q2-sample-classifier was called for Random Forest analysis and nested stratified cross test. The dominant species in each of the three quarters were determined by the number of sequence reads (species with more than 10,000 quarterly sequence reads were selected as the dominant species). Simultaneously, we analyzed the results of species annotation and classified fish to order, family, genus, and species levels. Finally, the eDNA data were compared with those of traditional methods for analysis and discussion.

2.3.4. Structure Division in the Community

Alpha and beta diversity indices were employed to measure species diversity within and across habitats, respectively, to thoroughly assess their total diversity [25,26]. Alpha diversity, which is within-habitat diversity, is used as a measure of species richness, diversity, and evenness in a locally homogeneous environment. Specifically, this study examined the alpha diversity of biological communities by characterizing richness by Chao1 [27] and observed species indices, Shannon [28] diversity and Simpson [29] indices; evolutionary-based diversity by Faith's PD [30] index; evenness by Pielou's [31] evenness index; and coverage by Good's coverage index [32].

We ran "qiime diversity alpha-rarefaction" using the unleveled ASV table with the parameters "-p-steps 10 -p-minute-depth 10 -p-iterations 10;" i.e., the minimum sampling level was 10, and the parameter "-p-max-depth" was set to 95% of the lowest sequencing depth in the entire sample, followed by the selection of 10 depth results randomly between this depth and the minimum depth, and the alpha division index. The average of the scores at the maximum leveling depth was chosen as the alpha diversity index.

To define the distribution of a sample at a particular distance scale, the non-metric multidimensional scaling (NMDS) analysis was used to decompose the sample distance matrix. The NMDS analysis did not rely on the computation of feature roots and feature vectors, but rather on ranking the distances between the samples to approximate, as closely as possible, the comparable distances in terms of closeness. Therefore, NMDS analysis did not affect the numerical values of sample distances and only evaluated the size connections between samples; consequently, the ranking conclusions may be more stable for complex data sets. A lower stress value indicates a more reliable NMDS analysis, and it is widely accepted that NMDS results are more reliable when this value is less than 0.02 [33]. Using the leveled ASV table, the "qiime diversity core-metrics-phylogenetic" command was invoked according to the presence or absence of the tree file; one distance matrices, Bray–Curtis, were calculated; the PCoA distance matrices were analyzed; and a QZV file was outputted. An NMDS analysis was performed on the Bray–Curtis distance matrix acquired from this study using R language scripts, and two-dimensional ordination plots were used to illustrate variations in fish community composition. The reason why Bray-Curtis distance is selected is that Bray-Curtis distance uses a weighted calculation method, that is, it calculates the sum of the absolute values of each species abundance difference between two samples and the ratio of their total abundance to obtain relevant experimental results, which considers not only the existence of species, but also the species abundance difference. Jaccard distance mainly calculates the proportion of non-shared species in all species between two samples, emphasizing the existence of species, but not the abundance of species [34]. However, the difference analysis of seasonal changes of fish communities needs to be based on abundance, covering all species of the two samples to be analyzed.

2.3.5. Seasonal Variation of Fish

The analysis of the seasonal changes of fish requires that all samples be compared at the same sequencing depth, so some transformation processing was required so that the rarefaction method could be used. This method randomly extracts a certain number of sequences from each sample to reach a uniform depth [35,36], which facilitates prediction of the observed ASVs and their relative abundance at this sequencing depth; this process is also called extraction leveling. The qiime feature-table Rarefy function was used, and the extraction depth was set as 95% of the minimum sample sequence quantity. To further compare the difference in species composition between samples in different seasons and show the distribution trend of species abundance in each sample, we used the abundance data of species to draw a heatmap for species composition analysis. The horizontal and vertical coordinates of the heatmap were arranged based on the collection time of the samples. The analysis software used included R and the heatmap package. For analysis, R script was used to calculate the clustering results of each sample and each classification unit, which were presented in the form of interactive graphs. Data analysis and heat map drawing were carried out on the Personalbio cloud platform.

*2.4. Conventional Analysis Method*

The fish caught were identified to species, the weight and tail number of each fish were recorded, and the average catch per unit time was expressed as the catch per unit time (kg/h) and the catch tail per unit time (tail/h) [37]. The fish in Laizhou Bay and the Yellow River Estuary were divided based on suitable temperatures, including warm-water species (marine organisms with a reproductive temperature range higher than 20 °C), temperate-water species (growth and reproduction of fish in the range of 4 °C to 20 °C), and cold-water species (growth and reproductive temperature range for fish below 10 °C) [38]. Dominant fish species were determined by the Relative Importance Index (IRI) [39]:

$$\text{IRI} = (\text{N} + \text{W}) \times \text{F}.$$

In the formula, N is the percentage of the tails of a certain class as a percentage of the total catch tails; W is a percentage of the total catch of a certain class; and F indicates the percentage of stations in which a certain fish appears as a percentage of the total number of stations. Fish with IRI values greater than 500 were considered dominant species, 500–100 were common species, 100–10 were considered general species, and 10 or lower were considered rare species [40]. In this paper, dominant and common species were considered important species components in fish communities.

## 3. Results

*3.1. Fish Species*

3.1.1. eDNA Outcomes

Based on the monitoring results of environmental DNA technology of this study and the quality control of the high-throughput sequencing library, a rough species annotation yielded 77 species of fish that belonged to 62 genera of 32 orders and 49 families. Based on the field conditions in the Bohai Sea and Laizhou Bay and by taking the anthropogenic pollution into account, the annotation results were screened and 47 species (Table 2) of fish were obtained. Among them, 32 species were found in the autumn of 2020, and 18 and 36 were found in the spring and summer of 2021, respectively. As shown in Table 3, autumn had one dominant species (*Cynoglossus joyneri*) and five common species (*Sebastes schlegelii*, *Thryssa kammalensis*, *Larimichthys polyactis*, *Platycephalus indicus*, and *Scomberomorus niphonius*). There were 10 dominant species in spring 2021. There were 14 dominant species in summer 2021. In autumn 2020, and spring and summer 2021, the dominant species accounted for 18.8%, 55.6%, and 38.9%of the total population, respectively. In Laizhou Bay and Yellow River Estuary, there were 20 species of warm-water fish, 21 species of temperate-water fish, and six species of cold-water fish, which represented

41.6%, 45.5%, and 13.0%, respectively, of the total fish species. Temperate-water species were the most abundant, followed by warm-water and cold-water species. The number of species found in warm water increased from spring to autumn. It can be seen that temperate-water species and warm-water species dominated. We counted the number and proportion of temperate-water species and warm-water species in the three seasons, and the results are as follows: there were 31 species in autumn 2020, accounting for 96.9% of the total fish population in autumn, 13 species in the spring of 2021, accounting for 81.3% of the total fish population in spring, and 30 species in the summer of 2021, accounting for 83.3% of the total fish population in summer.

**Table 2.** Sequence annotation result ("+" indicates existence, common name for a vacancy is a name that has not yet been normalized).

| Species | Common Name | Autumn | Spring | Summer |
|---|---|---|---|---|
| *Acanthogobius hasta* | Asian fresh water goby | | | + |
| *Acanthopagrus schlegelii* | | + | | |
| *Amblychaeturichthys hexanema* | Blackhead Seabream | | | + |
| *callionymus beniteguri* | Pinkgray goby | + | | + |
| *Chaeturichthys stigmatias* | | + | + | + |
| *Coilia mystus* | | + | | + |
| *Collichthys lucidus* | Tapertail anchovy | | | + |
| *Cryptocentrus filifer* | | + | | + |
| *Ctenotrypauchen chinensis Steindachner* | light maigre | + | | |
| *Cynoglossus joyneri* | | | | + |
| *Cynoglossus robustus* | | | | + |
| *Cynoglossus semilaevis* | | + | | + |
| *Enedriasfangi* | robust tonguefish | + | | |
| *Engraulis japonicus* | | + | + | + |
| *Harpadon nehereus* | Tongue Sole | + | + | + |
| *Hexagrammos otakii* | Fang's gunnel | | | + |
| *Ilisha elongata* | | | + | + |
| *Jaydia lineata* | Japanese anchovy | + | + | + |
| *Johnius belangerii* | | + | | |
| *Johnius grypotus* | Bombay duck | + | | |
| *Kareius bicoloratus* | The Otaki six-liner | + | + | + |
| *Konosirus punctatus* | Ilisha elongata | + | | + |
| *Larimichthys crocea* | Verticalstriped Cardinalfish | + | + | + |
| *Larimichthys polyactis* | | + | | + |
| *Lateolabrax japonicus* | Belanger's croaker | + | | |
| *Lateolabrax maculatus* | | + | + | + |
| *Liparis tanakae* | | | | + |
| *Miichthys miiuy* | Stone flounder | + | | + |
| *Mugil cephalus* | | + | + | |
| *Muraenesox bagio* | Dotted Gizzard Shad | | + | |
| *Muraenesox cinereus* | | + | | |
| *Pennahia argentata* | Large yellow croaker | + | | + |
| *Platycephalus indicus* | | + | + | + |
| *Pleuronichthys cornutus* | Little Yellow Croaker | | + | + |
| *Protosalanx hyalocranius* | | + | + | + |
| *Rhinogobius similis* | Japaneseseaperch | | | + |
| *Sardinella zunasi* | Spotted Seabass | + | + | + |
| *Scomber japonicus* | | + | | + |
| *Scomberomorus niphonius* | | + | + | + |

**Table 2.** *Cont.*

| Species | Common Name | Autumn | Spring | Summer |
|---|---|:---:|:---:|:---:|
| *Sebastes schlegelii* | Brown Croaker | + | | + |
| *Sebastiscus marmoratus* | | + | | + |
| *Setipinna taty* | Sea Mullet | | + | + |
| *Strongylura anastomella* | | | | + |
| *Synechogobius hasta* | Common pike conger | + | + | |
| *Terapon jarbua* | | | | + |
| *Thryssa kammalensis* | daggertooth pike conger | + | | + |
| *Zoarces gillii* | | | + | |

**Table 3.** Dominant species in each season.

| Season | Dominant Species |
|---|---|
| Autumn 2020 | *Cynoglossus joyneri* |
| Spring 2021 | *Scomberomorus niphonius, Ctenogobius giurinus, Larimichthys polyactis, Platycephalus indicus, Thryssa kammalensis, Kareius bicoloratus, Cynoglossus joyneri, Lateolabrax maculatus, Hexagrammos otakii,* and *Engraulis japonicus* |
| Summer 2021 | *Sardinella zunasi, Scomberomorus niphonius, Thryssa kammalensis, Engraulis japonicus, Platycephalus indicus, Larimichthys polyactis, Cynoglossus semilaevis, Hexagrammos otakii, Cynoglossus joyneri, Chaeturichthys stigmatias, Lateolabrax maculatus, Protosalanx hyalocranius, Konosirus punctatus,* and *Acanthogobius hasta* |

The Random Forest algorithm revealed the species diversity of several samples (Figure 2). The relevance of species to the models declined in descending order of abundance, with the species at the top serving as an indicator of population differences.

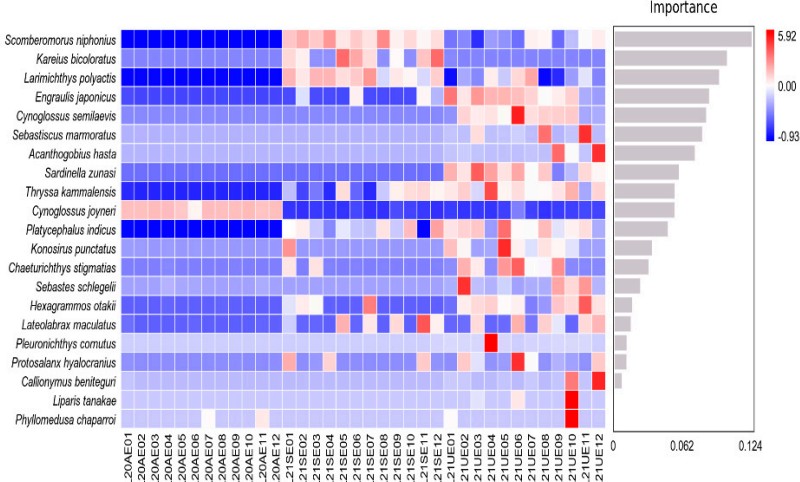

**Figure 2.** Scatter diagram of random forest analysis. The horizontal coordinates of the bars in the figure represent the importance scoring values of lifeforms to the classification algorithm, and the lateral coordinates are the names of taxonomic units at the species identification; the heatmap demonstrates the variation of an abundant supply of these lifeforms across samples/groups. The relevance of organisms to the framework declines from up to down; organisms at the top of the value scale might be regarded as indicator species for inter-group disparities.

3.1.2. Conventional Resource Survey Outcomes

The identification, counting, and weighing of species captured in ground cages, gill-nets, and trawl nets yielded 11 fish species over the course of two seasons (Table 4). *Sebastes schlegelii, Hexagrammos otakii,* and *Acanthogobius hasta* were present in both seasons, and all

three species are warm-water fish. Based on the catch amount, nine species were discovered in autumn 2020. According to IRI (Table 5), the dominant species were determined, among which *Sebastes schlegelii* and *Synechogobius ommaturus* were the absolute dominant species, and *Lateolabrax japonicus*, *Cynoglossus semilaevis*, and *Pleuronichthys cornutus* were the common species. Five species were found in spring 2021, among which *Sebastes schlegelii*, *Hexagrammos otakii*, *Acanthogobius hasta*, and *Ctenogobius giurinus* were the absolute dominant species, and *Enedras fangi* was a common species. As determined by the classification of suitable temperature types, only *Enedras fangi* was classified as a cold-water fish, whereas the other four species were classified as warm-water fish. As defined by the classification of ecological types, only *Lateolabrax japonicus* was considered an upper-layer fish, whereas the others live in the bottom layer and were considered lower-layer fish. In both seasons, *Sebastes schlegelii* was the dominant species.

**Table 4.** Information on fish obtained by conventional survey techniques ("+" for presence).

| Species | Autumn 2020 | Number | Spring 2021 | Number |
|---|---|---|---|---|
| *Sebastes schlegelii* | + | 53 | + | 3 |
| *Lateolabrax japonicus* | + | 10 | | |
| *Hexagrammos otakii* | + | 3 | + | 6 |
| *Synechogobius ommaturus* | + | 16 | | |
| *Chaemrichthys stigmatias Richardson* | + | 1 | + | 20 |
| *Sebastiscus marmoratus* | + | 1 | | |
| *Ctenogobiusgiurinus* | | | + | 11 |
| *EnedrasfangiWangetWang* | | | + | 3 |
| *Cynoglossus semilaevis* | + | 23 | | |
| *Pleuronichthys cornutus* | + | 6 | | |
| *Platichthys bicoloratus* | + | 3 | | |

**Table 5.** IRI value of fish in each season (W and N represent the percentages of fish weight and catches in the total catch weight and catches in each season respectively).

| Season | Species | W/% | N/% | IRI |
|---|---|---|---|---|
| Autumn 2020 | *Sebastes schlegelii* | 78 | 45.7 | 5715 |
| | *Lateolabrax japonicus* | 5.4 | 8.6 | 322 |
| | *Hexagrammos otakii* | 0.8 | 2.6 | 78 |
| | *Synechogobius ommaturus* | 6.7 | 13.8 | 1261 |
| | *Acanthogobiushasta* | 0.04 | 0.86 | 7 |
| | *Sebastiscus marmoratus* | 0.04 | 0.86 | 7 |
| | *Cynoglossus semilaevis* | 6 | 19.8 | 459 |
| | *Pleuronichthys cornutus* | 1.8 | 5.2 | 108 |
| | *Platichthys bicoloratus* | 1.1 | 2.6 | 57 |
| Spring 2021 | *Sebastes schlegelii* | 53 | 7 | 3426 |
| | *Hexagrammos otakii* | 30 | 14 | 2512 |
| | *Acanthogobiushasta* | 8.4 | 46.5 | 1175 |
| | *Ctenogobiusgiurinus* | 8 | 25.6 | 1441 |
| | *Enedrasfangi* | 0.54 | 7 | 108 |

### 3.2. Fish Community Diversity

As illustrated in Figure 3, there were substantial changes in the fish composition of Laizhou Bay for the three seasons examined in the study, with the maximum adequate supply in summer 2021 and the lowest in autumn 2020. Diversity was greatest in summer 2021 and spring 2021, and was practically similar in both seasons, with the lowest diversity in autumn 2020. Uniformity was likewise essentially comparable and strongest in spring and summer 2021, and lowest in autumn 2020. However, coverage was greatest in autumn 2020 and spring 2021, followed by summer 2021.

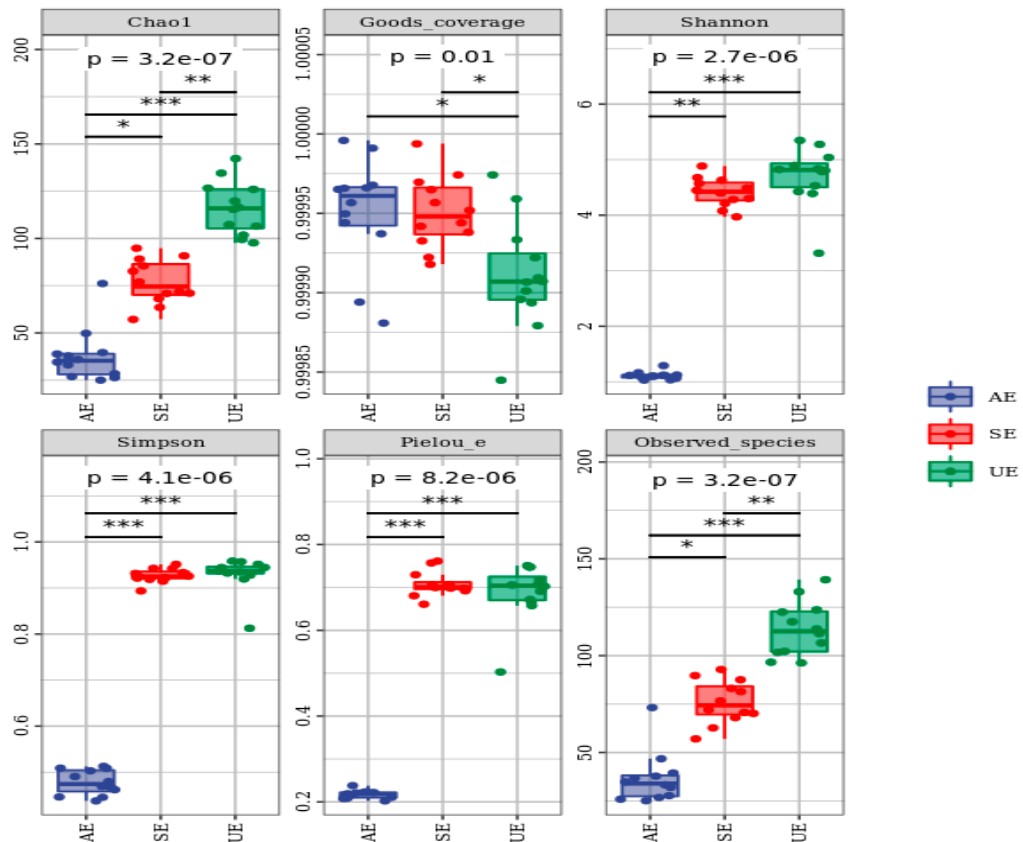

**Figure 3.** Seasonal difference based on the Alpha index. "*" represents *p* < 0.05; "**" represents *p* < 0.01, and "***"represents *p* < 0.001. Each panel corresponds to an alpha diversity index, identified by a gray area at the top. In each Panel, the abscissa is the grouping label and the ordinate is the value of the corresponding alpha diversity index; these three colors represent three reasons. The value under the diversity index label is the *p* value of the Kruskal–Wallis test. (Subgroup AE: Autumn of 2020. Subgroup SE: in the spring of 2021. Subgroup UE: the summer of 2021).

The seasonal combination mode was illustrated by a 2D ranking plot according to the NMDS analysis (Figure 4). The data demonstrated that there was closer separation of the two locations in spring and summer 2021, which showed that the variations in community makeup between the two seasons were modest. The considerable separation between them and the autumn 2020 seasonal community showed that there was major variation in community composition.

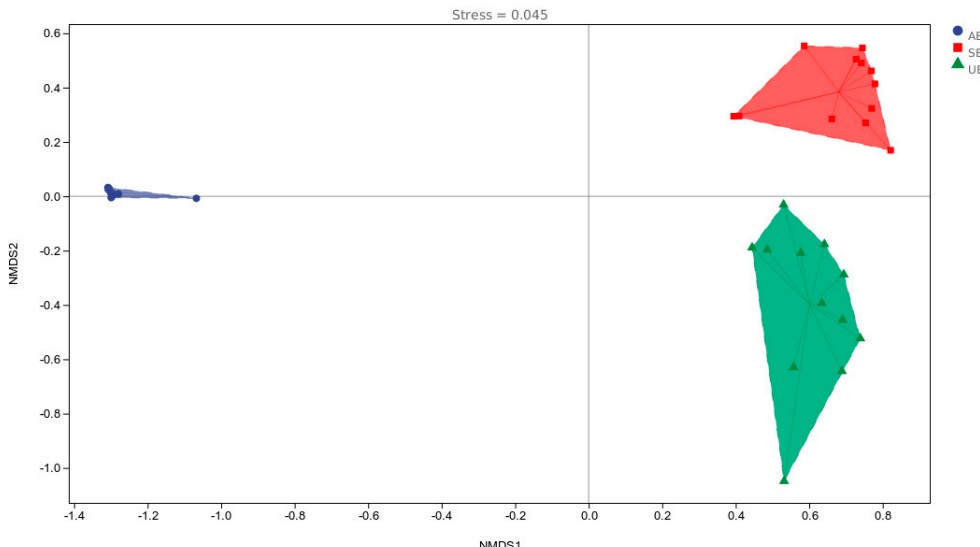

**Figure 4.** Nonmetric multidimensional scaling (NMDS) analysis of a two-dimensional ordered graph. Each point in the diagram represents a sample, and different colors represent different seasons (groups). It can be approximated that the closer (farther) the distance between two points, the smaller (larger) the difference in microbial communities between the two samples. (Subgroup AE: Autumn of 2020. Subgroup SE: in the spring of 2021. Subgroup UE: the summer of 2021).

*3.3. Seasonal Variation*

As shown in Figure 5, the structural composition of fish between the three seasons significantly varied. The lowest diversity occurred in spring, the lowest species richness occurred in autumn, and the highest species richness and diversity occurred in summer.

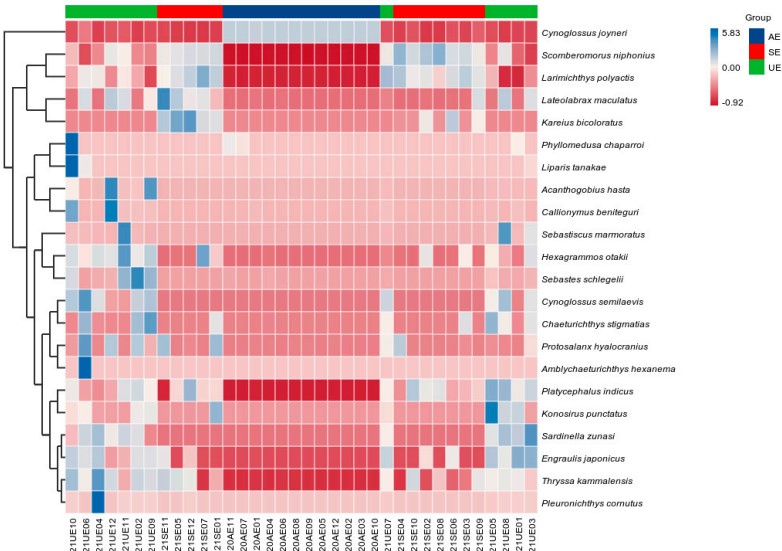

**Figure 5.** In the figure, samples are clustered as per distance metric (default clustering method) of the organism mixture information for UPGMA and are sorted based on the clustering results; nevertheless, they are arranged according to the default order of samples/groups. Species are clustered by default, i.e., UPGMA network topologies based on the Pearson coefficient of correlation matrix of their constituent information (default clustering algorithm), and ranked based on the clustering results. However, species are ranked based on their average abundant supply in the sample/group. (Subgroup AE: Autumn of 2020. Subgroup SE: in the spring of 2021. Subgroup UE: the summer of 2021).

## 4. Discussion

### 4.1. Fish Diversity and Annual Variation in Laizhou Bay

The waterways of Laizhou Bay and the Yellow River Estuary are important sites for the growth and development of various aquatic economic organisms, and fish are an important part of the biological community in the region. In this research, 77 fish species were found based on eDNA information, which is 67.5% of the number of fish species that were collected in 1982–1985 by survey techniques including ground cages, gillnets, and trawls. Economic species such as *Cynoglossus joyneri*, *Engraulis japonicus*, and *Thryssa kammalensis* increased in proportion based on the information obtained from eDNA. According to a study published by Jin et al. (2014), *Chaeturichthys stigmatias* was the most dominant species in all three seasons surveyed. The dominant fish species in Laizhou Bay and the Yellow River Delta have dramatically changed since the 1950s. The dominant species in 1959 were *Trichiurus lepturus* and *Larimichthys polyactis*. In 1982–1985, the dominant species were *Setipinna taty*, *Collichthys niveatus*, *Engraulis japonicus*, and *Larimichthys polyactis*. In 1992–1993, *Engraulis japonicus* was the absolute dominant species. In 1998, *Thryssa kammalensis* and *Setipinna taty* were more dominant; *Engraulis japonicus* was only a dominant species in spring; and *Setipinna taty* was the dominant species in spring, summer, and autumn [41]. According to the eDNA analysis conducted in this research, the following were the dominant species identified in Laizhou Bay during the three seasons surveyed: *Cynoglossus joyneri*, *Scomberomorus niphonius*, *Larimichthys polyactis*, *Thryssa kammalensis*, *Sardinella zunasi*, *Platycephalus indicus*, *Engraulis japonicus*, *Hexagrammos otakii*, *Cynoglossus semilaevis*, *Lateolabrax maculatus*, *Kareius bicoloratus*, *Chaeturichthys stigmatias*, *Protosalanx hyalocranius*, *Konosirus punctatus*, *Konosirus punctatus*, *Acanthogobius hasta*, *Sebastiscus marmoratus*, and *Sebastes schlegelii*. Four species were the absolute dominant species in the three seasons and were all small fishes: *Cynoglossus joyneri*, *Scomberomorus niphonius*, *Larimichthys polyactis*, and *Thryssa kammalensis*.

According to the data obtained by the traditional resource survey method, the number of fish species in Laizhou Bay decreased annually (Figure 6). This data is consistent with the trend of fish resources survey in Laizhou Bay in recent years. Compared with the results of Wang et al. (2011), the eDNA results showed an increase in increase in fish populations, which may be because eDNA technology performs species annotation at the molecular level and can monitor some species that are not easily detected using conventional techniques. In contrast, implementation of synthetic preservation methods in recent years, such as artificial stocks and periodic closure, have aided the ecological recovery of afflicted regions in Laizhou Bay [42]. Nevertheless, the decrease in biomass and decline of fishery resources continue to be a severe concern, and further study on fish population composition and decline is required to guide future ecological restoration efforts.

Presently, the principal dominant populations of fishery resources in Laizhou Bay and the Yellow River Estuary have changed as follows: *Setipinna taty* to *Engraulis japonicus* to *Thryssa kammalensis* to *Cynoglossus joyneri* towards more miniaturization (Table 6). Fish community structure may have changed as a result of the decline in ecosystem resilience and biome integrity caused by overfishing and environmental degradation, and the decline in ecosystem stability [43]. Over the past few years, selective commercial overfishing has caused a rapid reduction in the number of economic fish in these waters that exceeded their regeneration capability and resulted in a dramatic decline in their numbers. Because of the reduction in large economic fish and the low number of migratory fish arriving in the waters in spring and summer, small fish, especially demersal fish, no longer have as many chances to be preyed upon, which leads to an increase in their numbers [1]. In addition, the land-based pollution caused by coastal economic development, the regulation of runoff by the construction of water conservation projects in the upper Yellow River reaches, and the destruction of the physical environment of the near-shore habitat from construction of aquaculture projects have all had significant effects on fishery resources [44]. For example, the content of cadmium, a heavy metal, in the tissue of fish in Laizhou Bay waters is the highest among China's coastal waters, and the contents of lead, mercury, and chromium,

which are also heavy metals, are among the top five [45]. Through the data obtained by eDNA, we identified the fish in Laizhou Bay and used the sequence reads to determine the dominant species in each season; this provides a new idea for the identification of dominant species. Comparing the voyage survey data of different years and the same quarter revealed the annual changes of fish in Laizhou Bay, which provides more complete data regarding the changes of fish populations, and this information can be used to protect fish stocks in Laizhou Bay.

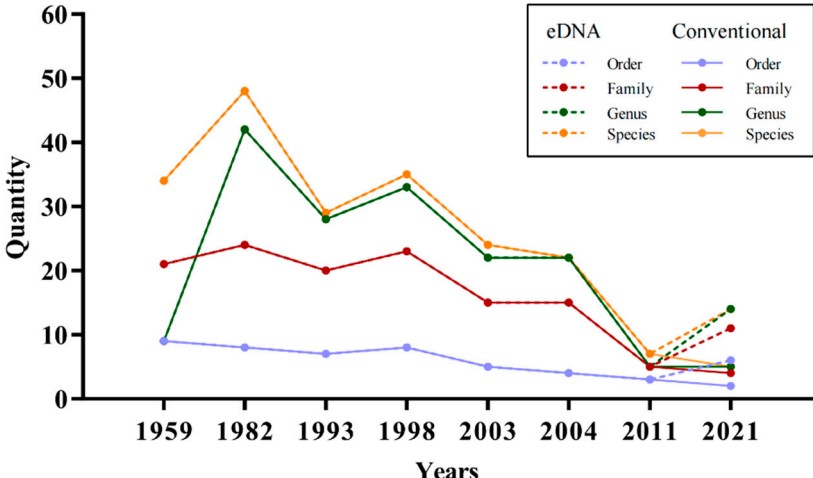

**Figure 6.** An inter-annual variation chart of fish communities in Laizhou Bay (The solid line represents the traditional survey method, and the dashed line represents the results of the eDNA). The sources of data are the same as in Table 6, the information for each year comes from the spring voyage in May, and the information for 2021 comes from the eDNA information of this research in spring 2021 and information from traditional survey methods).

**Table 6.** Species composition in different years. (Both the surveys are conducted in the spring of each year.).

| Year | Species Number | Technique |
|---|---|---|
| 1959 [41] | *Trichiurus lepturus, Larimichthys polyactis, Cynoglossus semilaevis, Pennahia argentata* | Conventional investigation technique |
| 1982 [46] | *Setipinna tenuifilis, Engraulis japonicus, Nibeaalbiflora, Larimichthys polyactis, Lateolabraxjaponicus;* | Conventional investigation technique |
| 1993 [47] | *Engraulis japonicus, Thryssa kammalensis, Setipinna tenuifilis, Konosirus punctatus* | Conventional investigation technique |
| 1998 [48] | *Thryssa kammalensis, Eneraulisiaponicus, Setipinna tenuifilis, Konosirus punctatus:* | Conventional investigation technique |
| 2003 [49] | *Thryssa kammalensis, Setipinna tenuifilis, Larimichthys polyactis, Pampus argenteus* | Conventional investigation technique |
| 2004 [49] | *Setipinna tenuifilis, Liparis tanakae, Enedriasfangi, Thryssa kammalensis* | Conventional investigation technique |
| 2011 [50] | *Symechogobius hasta, callionymus beniteguri, Cynoglossus joyneri, Chaeturichthys stigmatias, Enedriasfangi* | Conventional investigation technique |
| 2021 | *Scomberomorus niphonius, Larimichthys polyactis, Platycephalus indicus, Thryssa kammalensis, Kareius bicoloratus, Cynoglossus joyneri, Lateolabrax maculatus, Hexagrammos otakii, Engraulis japonicus* | eDNA (outcome of this research) |
| 2021 | *Sebastes schlegelii, Hexagrammos otakii, Acanthogobiushasta and Ctenogobiusgiurinus* | Conventional investigation technique (outcome of this research) |

*4.2. Seasonal Variation of Fish in Laizhou Bay*

We analyzed the differences between quarters based on serial quantity, and found that autumn 2020 had significantly less diversity than the two other quarters. For additional analysis, we performed diversity and seasonal analyses, and found that spring had the lowest diversity, autumn had the lowest species richness, and summer had the highest species richness and diversity.

Spring and autumn had the lowest diversity and richness of all three seasons, whereas summer had the highest diversity and richness. The differences in fish temperature suitability and habitat type may explain this result. In a 2016–2017 study, the compositions of temperate-water, warm-water, and cold-water species in the overall wide range of aquatic lifeforms in Laizhou Bay, which were distinguished by thermoregulation type, were 59.6%, 23.1%, and 17.7%, respectively; it was roughly estimated that, based on the fish thermoregulation type, the number of temperature-water species was the highest, which accounted for 60% of the total number of species, whereas the number of warm-water species accounted for 23.1% of the total number of species [51]. Both warm- and cold-water species accounted for 20% of the total number of species. Based on the type of fish habitat, the shallow water bottom layer of the continental shelf contained the most fish species, followed by the shallow water middle layer of the continental shelf. In Laizhou Bay, the number of warm-water species was highest in all seasons, and their biomass was predominant. The number of shallow bottom fish species in the continental shelf was the highest in all seasons, and summer had the highest biomass, which was replaced by shallow bottom fish in winter and autumn. Compared with previous research, the number of fish species in Laizhou Bay has significantly decreased, but the composition of temperature-adapted species has not significantly changed.

The number of temperature-water and warm-water species gradually increased from spring to autumn. This is partly because Laizhou Bay and the Yellow River Estuary have a more suitable environment and temperature for biological growth in summer, and an abundant supply of summer bait compared with autumn, winter, and spring. However, it is also due to the spawning migration of Yellow Sea fishes from their overwintering grounds to the coast of the Bohai Sea beginning each spring [51,52]. For example, *Larimichthys polyactis* begin spawning in June in various harbors of the Bohai Sea, the northern coast of the Yellow Sea, and Haizhou Harbor, and *Larimichthys polyactis* inhabiting the Bohai Sea seek bait in the middle of the Bohai Sea from September to November before migrating to the overwintering grounds around the Chengshan Cape after November [53]. Therefore, the fish populations migrating from the Yellow Sea into the waters of Laizhou Bay and the Yellow River Estuary in the spring are still relatively small, whereas fish resources in autumn primarily consist of supplementary populations, and bait organisms are more abundant in summer.

The dominance of *Konosirus punctatus*, *Engraulis japonicus*, and *Larimichthys polyactis* increases as fish populations rapidly expand. Previous research has indicated that water temperature, depth, and salt content have a substantial effect on fish communities. The spring and summer 2021 fish assemblages were extremely similar, which is possibly because of the similar temperature conditions in both seasons leading to identical temperature-adapted fish [54]. The eDNA data clearly showed the changes of fish in Laizhou Bay over three seasons and provided a molecular-level technical means for monitoring seasonal changes of fish in Laizhou Bay.

*4.3. eDNA vs. Conventional Resource Survey Methodologies*

Using eDNA and a conventional resource survey, we analyzed the fish distribution and community structure in the marine grassland region of Laizhou Bay. The conventional resource survey yielded 11 fish species in two seasons, whereas the eDNA survey yielded 47 fish species in three seasons (Table 7). Comparative analysis of the two seasons in which the two techniques were concurrently employed. Initially, ground cages, gillnets, and trawls were used as conventional techniques for fish diversity surveys, and the experimental

results were analyzed in terms of catch, dominant length of main catch, number of tails, and catch of economic species. In autumn 2020 and spring 2021, 11 fish species were discovered based on the conventional resource survey information and were generally consistent with the eDNA results, with the exception of *Enedras fangi*, which appeared in the ground cage survey but not in the eDNA information. In addition, eDNA was compared with the dominant species obtained by conventional survey techniques (Table 7). Therefore, it can be seen that both Sebastes schlegelii and Hexagrammos otakii dominated in spring, but there were no common dominant species in autumn.

**Table 7.** Comparison of eDNA and conventional investigation technique.

| Technique | Number of Species | Dominant Species in Autumn 2020 | Dominant Species in Spring 2021 |
|---|---|---|---|
| eDNA | 47 | *Cynoglossus joyneri* | *Scomberomorus niphonius, Ctenogobius giurinus, Larimichthys polyactis, Platycephalus indicus, Thryssa kammalensis, Kareius bicoloratus, Cynoglossus joyneri, Lateolabrax maculatus, Hexagrammos otakii,* and *Engraulis japonicus* |
| Conventional investigation technique | 11 | *Sebastes schlegelii* and *Synechogobius ommaturus* | *Sebastes schlegelii, Hexagrammos otakii, Acanthogobius hasta,* and *Ctenogobius giurinus* |
| | 10 | / | *Sebastes schlegelii, Hexagrammos otakii* |

One of the reasons for the discrepancy between the eDNA information and the conventional survey is that the conventional survey primarily used catch and IRI to ascertain the dominant species, whereas we used a large number of sequence reads for the eDNA research; this resulted in differences between the detected dominant species [22,55]. Compared with conventional fish stock surveys, we devised a new technique for information screening and analysis. The ground cage, which is a conventional survey technique, provides a very visual representation of information; however, because its mesh size is fixed, many small-bodied fish are not easily captured, which results in incomplete information. Results from Ren et al. (2018) indicated that both the number and species escape rate of catches increased with mesh size, with the species escape rate of 40–60 mm mesh reaching approximately 80%, the tail escape rate exceeding 90%, and the mass escape rate reaching more than 50%. This finding indicated that 40–60 mm mesh had an obvious release effect, and the proportion of economic species caught was positively correlated with mesh size. As a biological monitoring technology at the molecular level, eDNA technology provides more comprehensive information [56].

However, despite the scientific maturity of eDNA techniques, several factors affect eDNA extraction and results, such as the source of eDNA, shedding rate, degradation rate, translocation, and sedimentation [57]. Therefore, the presence of eDNA in the water column has a real-time, easy-to-affect biological relationship with the quantity of fish resources. Moreover, false positives caused by the influence of exogenous DNA from other waters cannot be excluded [58,59]. Because DNA can exist for a long time after it is shed from the organism, improper sample processing may result in incorrect inferences. Concerns such as interference from closely related species, the activity cycle of fish, and the direction of current velocity should be considered when using eDNA to investigate fish biomass. These factors also contribute to the minor disparities between eDNA data and conventional survey techniques. The main species composition of this research tends to be consistent with the results of eDNA-based fish monitoring in Laizhou Bay, which further validated the reliability of the data. By comparing eDNA with conventional resource surveys, we further confirmed that eDNA can be used as a complementary tool to monitor fish in Laizhou Bay.

*4.4. eDNA as a Revolutionary Method for Analyzing Fish Divergence*

eDNA outcomes can be used to monitor both major commercial fish species in Laizhou Bay, such as *Lepturacanthus savala*, *Scomberomorus niphonius*, and *Larimichthys polyactis*, and also relatively rare species, such as *Lophius litulon* and *Scomber japonicus*, which exist in a small amount in the Bohai Sea and are bottom fish [60]. In 2020, the endangered species *Takifugu rubripes* was discovered in the Yangtze River Estuary [22]. This discovery is very beneficial for the preservation of fisheries in Laizhou Bay. Conventional survey approaches have a limited probability of locating threatened or protected species, especially if the organism is submerged in an aquatic habitat [61]. These studies indicate that eDNA technology is safe and that it is possible to harvest eDNA for fish monitoring in aquatic environments without harming the fish themselves. Therefore, eDNA analysis is a useful method for monitoring both endangered and relatively uncommon species.

Compared with our team's 2018 research, we enhanced the quantitative results by using the Personalbio cloud platform findings in this research [17]. The use of cloud platform analysis to combine quarterly samples and extract equation analysis decreased mistakes and other problems resulting from missing samples [62]. The results of this study also indicate that eDNA technology may be useful for evaluating the diversity, abundance, biomass, and seasonal geographic distribution of fish species in Laizhou Bay. Additionally, 12S rRNA segments were employed as amplification primers in our investigation. We employed the 12S rRNA primers provided by Miya et al. (2018) to study the Yangtze River Estuary using eDNA and achieved accurate identification results [17,19]. Compared with our other studies [2], this study expanded the investigation area.

**5. Conclusions**

This study is the first to use eDNA to evaluate fish diversity and seasonal variation in Laizhou Bay. In total, 47 fish species were identified; spring had the lowest diversity, autumn had the lowest species richness, and summer had the highest diversity and species abundance. Our results demonstrated that monitoring the diversity and seasonal changes of fish resources in Laizhou Bay was superior with eDNA technology compared with conventional monitoring techniques, and eDNA had obvious benefits. These results demonstrated that eDNA can be used as a source of information on space and time, to obtain DNA of species in environmental samples that can be used to infer taxonomic information and gene functions, to study the species diversity in various ecological environments, and to trace the origin of species development and change [63]. eDNA detection will play an increasingly important role in studies on environmental biodiversity and specific organisms [52]. However, additional studies are required to determine the biomass of target species because of the effect of numerous unknown environmental conditions [64]. This approach also has several disadvantages, including the impact of contaminated waterway samples and PCR contamination, and ambiguity. Consequently, eDNA technology may supplement conventional fish survey approaches and provide more thorough knowledge of the impact of human activities and other variables on the structure and diversity of fish communities. The data from this study show that eDNA is an effective tool for monitoring fish stocks in Laizhou Bay.

**Author Contributions:** Conceptualization, S.D.; Data curation, S.D. and H.Z.; Formal analysis, S.D.; Methodology, S.D.; Project administration, H.Z.; Resources, W.X. and H.Z.; Software, S.D.; Supervision, M.B., H.J., W.X. and H.Z.; Validation, S.D. and H.Z.; Writing—original draft, S.D.; Writing—review & editing, H.J. and H.Z. All authors have read and agreed to the published version of the manuscript.

**Funding:** The present work was supported by National Key Research and Development Project of China (2019YFD0902101) and Youth Innovation Promotion Association Chinese Academy of Sciences (2020211).

**Institutional Review Board Statement:** Not applicable.

**Informed Consent Statement:** Not applicable.

**Data Availability Statement:** All data or models generated or used during the study are available from the corresponding author by request.

**Acknowledgments:** We would like to thank the editor and all reviewers for the constructive comments on our present study. We would like to thank Yibang Wang for his help in revising the article and analyzing the data.

**Conflicts of Interest:** The authors declare no conflict of interest.

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
