# Peer review of "An Assessment of Seasonal Differences in Fish Populations in Laizhou Bay Using Environmental DNA and Conventional Resource Survey Techniques"

_fishes, doi:10.3390/fishes7050250_

Round 1

Reviewer 1 Report

The manuscript analyzed fish populations in Laizhou Bay using environmental DNA and conventional resource survey techniques for assessment of seasonal differences. However, there are some major and minor comments in the manuscript. The detailed remarks are suggested below.

The following is major point should be revised:

l  The paragraphs of the introduction explain about previous researches too much. The authors should rewrite an introduction to justify the purpose of this study. Also the authors should add the reasons why the authors examine the seasonal fluctuations using eDNA and the reasons why the authors compare the eDNA with conventional survey techniques.

l  The authors divided sampling sites into sub-areas based on different conditions: Area for the integration of wind power in pastures, Reef area, Surrounding reef area, and Control area. However, the authors did not include any further biodiversity analyses based on sub-areas. Therefore the authors should explain why the sub-areas are divided and biodiversity analyses are required accordingly In addition it is necessary to analyze biodiversity for each sub-area for purpose of comparison if possible.

l  In the materials and methods, the authors mentioned that collected 1-2L of water for eDNA sampling. The results are not affected by the quantity of sampled water?

l  The authors examined for assessment of seasonal differences in fish populations. However, the authors examined only three seasons spanning one year. It cannot be clearly explained whether the results are due to seasonal differences or not. To support the results, the authors should study for a long period, at least one year. Also, the authors should suggest which environmental factors such as water temperature, pH and DO, etc affect mostly the seasonal differences. If possible the authors may present PCA plot showing relation between environmental factors and species abundance and diversity.

l  In line 331, the authors analyzed species diversity using a random forest algorithm. The information on a random forest algorithm used should be in the Materials and methods.

The following is minor point should be revised:

l  Line 60-62: This reference did not based on eDNA. The authors should check the reference.

l  Line 205: Change “DaDA2” to “DADA2”

Reviewer 2 Report

I have now read through Dai et al. "An assessment of seasonal differences..."

I was very interested to read this study. The data are very interesting and the study is intriguing. I think this study can be impactful. However, I think it requires some work first. I'll identify some areas where I think the authors have work to do.

- the introduction is very thorough. Too thorough. I suggest it be narrowed in scope and perhaps reduced in length. eDNA metabarcoding is well enough established now that a paper such as this does not need to begin a literature review with microbial work in the 1980s. I think a more focused review of the use of eDNA metabarcoding in coastal marine environments would be more appropriate.

-Likewise, there are parts of the methods that offer unnecessary detail. For example, beginning on line 171, it is not necessary to go into details of how the commercial library prep kit works. Some of the step by step details could be replaced by appropriate references.

-line 231, "consulted books". Why are these books not cited here?

-line 246 "examined alpha diversity of microbial communities". this was a big red flag for me. Why are the authors making the mistake here of referring to "microbial communities"? This was not a singular error either. "microbial" is mentioned in other places as well, as for example in the figure 4 legend. This is a big mistake and it makes me wonder if sentences were lifted from other sources without careful scrutiny. I won't say that I detected plagiarism, because I did not evaluate for plagiarism, but encourage the authors and the editor to review carefully for the possibility of lifted sentences. How else would "microbial diversity" end up in this manuscript?

-The authors have a nice section in the methods describing a definition of Dominant fish species based on a relative importance index based on conventional catch data. I don't see this method applied to the conventional data in the results or mentioned in the discussion. However, lines starting at 314 report at length dominant species by season based on eDNA data. I don't think the authors provide a relative importance index based on eDNA. I think there is something very important missing here.

-Line 310 reports 77 species in table 2. However, there are only 47 species in the table. I did not understand lines 311-313 and how the authors went from 77 well characterized species (62 genera, 32 orders, 49 families) to 47 species. This is important and it did not make sense to me.

-In some places in the results the authors narrated long lists of species that are just challenging to read. Lines 315-323 are an example. The authors need to make better use of their tables to avoid these difficult narratives. Lines 428-432 are another example of text that just lists long lines of scientific names. These are really impossible lines to read and comprehend for anyone who is not intimately familiar with these species.

-I have a difficult time understanding what figure 4 is supposed to show me. It does not help that the legend references microbial communities.

-for all the figures that were generated probably in R, please figure out how to change "A", "B" and "C" to more useful group names. It is not sufficient to identify those names in the figure legends.

- In figure 3, why show every measure of alpha diversity possible. These different measures of alpha diversity were not important to the discussion. Pick the appropriate diversity metrics and report just those.

-I feel like an effective comparison of eDNA and conventional surveys was missing from this paper.

-Figure 6 provides interesting historical perspective on the fish community in this bay. The 2021 data are explained away as an artifact of the method used (line 444). So then why not show what this graph would look like if the conventional sampling data were included instead? You have the data, correct?

-line 501 - how do you know that the number of warm water species gradually increased from spring to autumn? You only have three data points. Can you more clearly define what you mean here by gradually?

Round 2

Reviewer 1 Report

The manuscript analyzed fish populations in Laizhou Bay using environmental DNA and conventional resource survey techniques for assessment of seasonal differences. In addition, the authors revised the manuscript to reflect the reviewer's comments. However, there are additional major and minor comments in the manuscript. The detailed remarks are suggested below.

The following is major point should be revised:

l  In the introduction (line 56-79), the authors wrote this paragraph to explain that "eDNA is an effective tool for monitoring of fish biodiversity". However, the sentences about eDNA monitoring of invasive species are not consistent with aim of this study. It is recommended to the authors remove line 62 to 67 (from "In the ~" to "~ decreased environmental interference")

l  The authors should explain the importance of Laizhou Bay (why fish biodiversity has to study in Laizhou Bay) in the introduction. Line 73 to 76 is not sufficient.

l  In figure 6, the eDNA graph (left) data until 2011 and the conventional resource survey graph (right) until 2011 are the same data. However, it may confuse the reader that the graphs in the manuscript are written with different data. The authors should present with other charts or tables the comparison of eDNA and conventional resource survey of 2011.

The following is minor point should be revised:

l  Please check for typos throughout the manuscript.

Reviewer 2 Report

This manuscript has really been substantially improved relative to the first submission. I think it is appropriate for publication. I have a couple suggestions for the authors:

1.    I think the authors rely on read counts in some of their diversity statistics and interpretations of dominant species. In the discussion there is a nice section on the kinds of factors that can influence eDNA prevalence and detectability of a species in this marine environment. Still, there has been a lot of debate in the literature on whether read counts can be interpreted as abundance and reflective of relative species biomass or if it is more appropriate to stick with presence/absence statistics with eDNA community data. Just as one example, the authors indicate on line 246 that they calculate both Jaccard (presence/absence) and Bray-Curtis (abundance) dissimilarity statistics. They indicate in the next lines that they constructed NMDS based on the Bray-Curtis matrix, but I don’t think they really justify why. I think it would be helpful in the methods for the authors to indicate which statistics they use are based on read counts and which are based on presence/absence, and to provide some justification for their use.

2.    Figure 6 was inserted in response to a comment I made about why not include the conventional data for the last data point in the graph. Thank you for adding that data. I’m not sure that two separate graphs are necessary here, since they are identical except for the last data points. Can you include a single figure, maybe figure 6b. And then have the eDNA data points plotted in the same graph, perhaps as a different symbol, to show what happens to the long term trend when eDNA data is included?

3.    Line 579 in the conclusions – is the number 77 intended, or should it be 47 species (per line 291)?

4.    I detected some small minor grammatical errors throughout that could be caught by another grammar review.
